# Investigation of the Rheological Properties of Zn-Ferrite/Perfluoropolyether Oil-Based Ferrofluids

**DOI:** 10.3390/nano11102653

**Published:** 2021-10-09

**Authors:** Fang Chen, Xiaobing Liu, Zhenggui Li, Shengnan Yan, Hao Fu, Zhaoqiang Yan

**Affiliations:** 1Key Laboratory of Fluid and Power Machinery, Ministry of Education, Xihua University, Chengdu 610039, China; chenfang@mail.xhu.edu.cn (F.C.); yanshn@mail.xhu.edu.cn (S.Y.); 2School of Physics, University of Electronic Science and Technology of China, Chengdu 611731, China; fuhao@uestc.edu.cn; 3Zigong Zhaoqiang Sealing Products Industrial Co., Ltd., Zigong 643000, China; yanzhaoqiang12@126.com

**Keywords:** ferrofluid, rheological property, viscosity, magnetic field

## Abstract

The rheological properties of ferrofluids are related to various applications, such as sealing and loudspeakers, and have therefore attracted widespread attention. However, the rheological properties and their influence on the mechanisms of perfluoropolyether oil (PFPE oil)-based ferrofluids are complicated and not clear. Here, a series of PFPE oil-based ferrofluids were synthesized via a chemical co-precipitation method, and their rheological properties were revealed, systematically. The results indicate that the prepared Zn-ferrite particles have an average size of 12.1 nm, within a range of 4–18 nm, and that the ferrofluids have excellent dispersion stability. The activity of the ferrofluids changes from Newtonian to non-Newtonian, then to solid-like with increasing *w* from 10 wt% to 45.5 wt%, owing to their variation in microstructures. The viscosity of the ferrofluids increases with increasing *M*_w_ (the molecular weight of base liquid PFPE oil polymer), attributed to the increase in entanglements between PFPE oil molecules. The magnetization temperature variation of Zn-ferrite nanoparticles and viscosity temperature variation of PFPE oil together contribute to the viscosity temperature change in ferrofluids. The viscosity of the ferrofluids basically remains unchanged when shear rate is above 50 s^−1^, with increasing magnetic field strength; however, it first increases and then levels off when the rate is under 10 s^−1^, revealing that the shear rate and magnetic field strength together affect viscosity. The viscosity and its alteration in Zn-ferrite/PFPE oil-based ferrofluids could be deduced through our work, which will be greatly significant in basic theoretical research and in various applications.

## 1. Introduction

Ferrofluid, a functional fluid material, prepared by dispersing magnetic nanoparticles coated with surfactants in base liquids [1], which are widely used in sealing, biomedicine, aerospace, etc. [2,3,4], owing to its having both mobility and magnetism under magnetic fields. The rheological properties of ferrofluids play a significant role in various applications [5,6], for example, fluids of low viscosity are needed in high speed seals, or the heat induced by friction will make sealing pressure reduce [7]. However, fluids of high viscosity are often needed in big gap seals [8]. Besides this, the viscosity and particle concentration of ferrofluids will both affect the performance of loudspeakers of various types [9,10]. The magneto-viscous effect of the ferrofluid can be regulated via an appropriate magnetic field strength in a hydraulic system [11]. It follows that the rheological properties of ferrofluids will decide product quality and affect their further applications [12]. 

The influence of various factors on the rheological properties of ferrofluids have been extensively studied in previous works. These factors can be divided into two categories: one category is various shear conditions (magnetic field strength, magnetic field direction, shear rate, temperature, etc.), which will affect the rheological properties of ferrofluids and make shear thinning [13] or shear thickening [14] occur. The viscosity of silicon oil-based ferrofluids decreases with rising temperature, and increases with increasing magnetic field strength; the formation and destruction of chains and droplets are the key factors affecting magnetoviscous properties [15,16]. The viscosity of Co_3_O_4_/paraffin-based ferrofluids reduces with increasing shear rate, and the particle-to-particle bonds are thought to be responsible for the shear-thinning effect [17]. The viscosity of kerosene-based ferrofluid was related to the strength and direction of the magnetic field, due to the quasiperiodic chains or columns formed [18,19]. Another category is the differences in ferrofluids (nanoparticle concentration, average particle size, particle size distribution, base liquid types, types of particles, etc.)—all these factors will have different effects on the rheological properties. It was reported that the viscosity of paraffin-based ferrofluid increases with increasing particle concentration of nickel; this behavior is related to particle interaction increases, due to the increase in nanoparticles concentration [20]. The viscosity of ferrofluid with smaller-size particles is larger than that with large-size particles—this is due to the large number density of smaller size particles in the fluid [21]. The magnetoviscous effects of the ferrofluids are related to the content of large particles [22]. The influence of large particles on the macroscopic dynamic properties of ferrofluids is related to the formation of internal heterogeneous structures as a result of their magnetic dipole interations with one another [23]. It is shown that an increase in the concentration of small particles leads, on the one hand, to a rise in phase transition temperature and, on the other hand, to an increase in the threshold concentration of large particles [24]. For larger particles, cluster sizes increase considerably under the effect of stronger dipoles and the number of single particles is reduced; smaller particles, which have weaker dipoles, stay separated [25]. The strong magnetoviscous effect for small shear rates has a physical origin in strong hydrodynamical perturbations that appear in the shear flowing ferrofluid due to the chains [26]. In addition, the carrier fluid viscosity greatly affects the rotational and oscillatory rheological properties of ferrofluids, due to the fact that the structures of ferrrofluids tend to be larger and more stable in a carrier fluid with larger viscosity [27,28]. The viscosity values of PAO-60-based fluids were considerably higher than those of PAO-30-based fluids under the same conditions, reflecting a viscosity difference between the two liquid bases [29]. 

Besides this, the effect of heterogeneous structure changes on the rheological properties of ferrofluids have also been researched experimentally and theoretically. The magnetoviscous effect is explained by the hindrance of free rotation of magnetic particles in a shear flow due to magnetic torque aligning the particle’s momentum along the magnetic field direction [30]. Various heterogeneous structures form in ferrofluids, such as chain-like, drop-like, closed rings and branched structures [31,32,33,34,35]. With increasing magnetic field strength, an alignment of the magnetic momentums of the particles along a common direction is enhanced, encouraging the formation of chain-like structures [36]. When the dimensionless characteristic energy of magnetic interaction between particles exceeds a certain critical value, bulk droplike aggregates, consisting of large numbers of particles, can occur in the system [37,38]. When the ferrofluid fills a thin gap placed into a normal magnetic field, the droplike structures can overlap with the gap [39]. It was confirmed that under low shear rates, the magnetoviscous effect is irreversible under a magnetic field, due to the fact that aggregated structures form with the distortion of magnetic drops after the magnetic field is eliminated [40]. When the nanoparticles’ concentration and the energy of their magnetic interactions exceed some threshold magnitudes, the majority of the particles are united in an entangled ferrofluids net [41]. 

Significantly, the possibility of heterogeneous structure formation is decided by interparticle interactions in ferrofluids, which is expressed by the value of the interaction parameter (*λ*′) [42], as it expresses the interaction between magnetic nanoparticles resulting from the balance of magnetic and thermal energies [43]. If *λ*′ < 1, the Brownian motion dominates in the ferrofluids, making it homogeneous with no field-induced structures. If *λ*′ exceeds unity, indicating the domination of magnetostatic particle interactions over the thermal motion *kT*, chain formation along the magnetic field direction is revealed [44]. *λ*′ = 8, an almost gel-like behavior in ferrofluids, occurs due to entangled chain structures [45]. The value of *λ*′ is affected by multiple factors, such as particle size, the magnetization of the magnetic material and the type of particles. It was reported that when the particle size is less than 16 nm, no chainlike structures are exibited in the ferrofluids under the influence of an applied magnetic field, and they are not able to produce a significant magnetoviscous effect [46]. It has been found that in the case of magnetite the particles should be larger than 12 nm to contribute to the formation of structures, whereas 6.5 nm cobalt particles are already able to form chain-like structures [36]. 

In addition, real magnetic fluids are practically always polydisperse and often contain ferromagnetic particles that are large enough to agglomerate and influence macroscopical properties of ferrofluids strongly. It was concluded that the biggest particles in real ferrofluids can play a principal role in the formation of rheological properties of these systems, in spite of their small concentration [26]. 

However, the above rheological properties and the influencing mechanisms of ferrofluids could not been applicable to various application conditions. A few related studies on the rheological properties of PFPE oil-based ferrofluid with superior physicochemical characteristics have been reported. It was revealed by W. M. Sun that the viscosity of PFPE oil-based ferrofluids increased with increasing particle concentration [47]. It was indicated by M. J. R. Cantow [48] and Z. K. Li [49] that the viscosity of PFPE oil-based ferrofluid decreased with rises in temperature, which is related to the viscosity temperature characteristics of the base liquid. The effects of particle concentration and temperature on the rheological properties of the ferrofluids have been reported. However, the application conditions of ferrofluids, such as temperature, magnetic field strength and sealing ion speed in industry are often complex; viscosity and its alteration in PFPE oil-based ferrofluids could not be predicted—moreover, the influence mechanism is not clear and needs to be further researched. 

Here, a series of PFPE oil-based ferrofluids with different particle concentrations (*w*) and molecular weights (*M*_w_) of base liquids were synthesized, and the influencing mechanisms of their rheological properties were revealed, systematically. Our results indicate that the activity of the fluids changes from Newtonian to Non-Newtonian, then to solid-like with increasing *w*. The viscosity temperature change of the fluids is contributed by the property of Zn-ferrite nanoparticles and PFPE oil. The shear rate and magnetic field strength will both affect the viscosity of the ferrofluids. Besides this, the ferrofluids with matched and controllable viscosity could be selected according to the results, and the feasibility of the ferrofluid could be estimated under various application conditions, such as in high-speed or big gap seals. Taken together, the findings on the rheological properties of Zn-ferrite/PFPE oil-based ferrofluids will provide the basis for theoretical investigation and will improve their application performance.

## 2. Materials and Methods

### 2.1. Preparation of Zn-Ferrite Nanoparticles and PFPE Oil-Based Ferrofluid

Briefly, ferric chloride (FeCl_3_∙6H_2_O), ferrous sulfate (FeSO_4_∙7H_2_O), zinc chloride (ZnCl_2_), sodium hydroxide (NaOH) and hydrochloric acid (HCl) were purchased from Alfa Aesar Chemical reagent Co., Ltd. (Shanghai, China) PFPE oil (F-(C_3_F_6_O)_n_-C_2_F_5_, *M*_w_ = 2400 g/moL, 3500 g/moL, 4600 g/moL, 7480 g/moL) and PFPE acids (C_3_F_7_O-(C_3_F_6_O)_n_-C_3_F_4_O_2_H, *M*_w_ = 3800 g/moL) were purchased from Shanghai ICAN Chemical S&T Co., Ltd. (Shanghai, China). The average length of 100 PFPE acid (*M*_w_ = 3800 g/moL) molecules could be approximately calculated by Materials Studio, and the value is ~3.6 nm, as revealed by the author [50]. All chemicals were of analytical grade and were used without any further purification.

In a typical procedure, the Zn-ferrite nanoparticles (Zn_*x*_Fe_3−*x*_O_4_, the ratio of Zn^2+^
*x* is 0.2) were synthesized via a chemical co-precipitation method by the following steps: First, FeCl_3_·6H_2_O (0.032 M, 8.66 g, 99%), FeSO_4_·7H_2_O (0.0128 M, 3.56 g, 99%) and ZnCl_2_ (0.0032 M, 0.435 g, 99.99%) were dissolved in ultrapure water (600 mL), respectively, at 70 °C by stirring. Next, NaOH (0.2 M, 8 g 99.99%) was added to the solution until the pH reached 12; meanwhile, mechanical agitation was carried out. After 30 mins, the pH of the solution was adjusted to be ~6.5 by adding HCl solution, and subsequently, PFPE acids (0.00079 M, 3 g, 99%) were added in to coat particles. One hour later, the particles were cleaned by ultra-pure water until the pH of solution reached ~7, and the particles were separated by a magnet. Next, the particles were dried in an oven for 10 h and then grinded to a powder-like state using a mortar. Finally, the nanoparticles were dispersed by mechanical agitation under 500 r/min for 10 h, and a series of ferrofluids with various particle concentrations (10 wt%, 20 wt%, 30 wt%, 40 wt%) and with PFPE oils of various *M*_w_ (2400 g/moL, 3500 g/moL, 4500 g/moL, 7480 g/moL) were synthesized, respectively. A similar method has been reported by author [50]—a schematic illustration of the preparation process of ferrofluid is presented in Figure 1.

### 2.2. Characterization Methods

X-ray diffraction (XRD) analysis was performed on a DX-2700X X-ray diffractometer (Haoyuan Instrument, Dan Dong, Liao Ning Province, China) equipped with Cu Kα (*λ* = 0.154 nm) radiation. Scans were made from 10° to 80° at the rate of 0.085 deg./s. Transmission electron microscopy (TEM: JEM-2000EX) (JEOL, Tokyo, Japan) was employed to study the micro-structure and selected area electron diffraction (SAED) of synthesized Zn-ferrite nanoparticles. The scanning electron microscopy (SEM) images was acquired on JSM-6700 M (JEOL, Tokyo, Japan) with an energy dispersive X-ray spectroscopy (EDX). A vibrating sample magnetometer (VSM, Lake Shore 7410) (Lake Shore, Columbus, Westerville, OH, USA) was used for the magnetic characterization of nanoparticles at an external magnetic field ranging from −2.5 × 10^4^ Oe to +2.5 × 10^4^ Oe at 25 °C. The magnetism temperature (*M*-*T*) of Zn-ferrite of nanoparticles was measured in the temperature range of 100–60 °C. The rheological properties of PFPE oil-based ferrofluids were investigated using a rotational rheometer (MCR 301, Anton-Paar GmbH, Graz, Austria), which generates an external magnetic field in a direction perpendicular to shear flow. A cone-plate system of non-magnetic metal with a diameter of 20 mm was used and the gap was set to be 0.5 mm. A magnetic field is applied in the range of 0–500 mT, the temperature range is set from 25 °C to 140 °C, and the shear rate is regulated from 0.01 s^−1^ to 100 s^−1^. 

## 3. Results

### 3.1. The Internal Structures and Magnetism of Zn-Ferrite Nanoparticles

The internal structures and magnetism of Zn-ferrite nanoparticles are shown in Figure 2. The diffraction peaks of particles correspond to the planes, such as (220), (311), (400), (422), (511), (440) and (533), indicating the spinel structure of the FCC phase (Figure 2a) [51]. No additional impurity phase was detected within the limit of observation in the XRD patterns. It can be seen that the particles had a nearly spherical morphology, the average size of nanoparticles was 12.1 nm and the distribution span was from 4 nm to 18 nm (Figure 2b,c). Besides this, some nanoparticles aggregated together. The polycrystalline diffraction rings of SAED patterns (Figure 2b) further showed the crystal characteristics of Zn-ferrite nanoparticles. In order to confirm the elements in the region of the SEM image, EDS was performed (Figure 2d), revealing that the synthesized particles consisted of Fe, Zn, and O elements. In Figure 2e, the coated Zn-ferrite nanoparticles had a saturation magnetization of 57.4 emu/g and coercivity of 19.3 Oe, having the characteristics of superparamagnetic-like materials. In Figure 2f, the saturation magnetization of Zn-ferrite nanoparticles decreased with rising temperature, and the Curie temperature *T*_c_ was ~475 °C. It could be deduced that the magnetization of Zn-ferrite ferrofluids decreases with rising temperature and will disappear at ~475 °C.

### 3.2. The Dispersion Stability of Ferrofluids

The long-term dispersion stability of ferrofluids is greatly important in various applications and research. The stability of ferrofluids can be characterized by an observation method—dynamic light scattering (DLS)—the centrifugation of which, is a simple and intuitive method for analyzing the stability of various ferrofluids. 

The repulsive forces contributed by the coated surfactant PFPE acids on Zn-ferrite nanoparticles were important in preventing interparticle aggregation, and were caused by van der Waals and dipolar interactions. All the synthesized PFPE oil-based ferrofluids were put in 50 mL centrifuge tubes and centrifuged under 5000 r/min for 2 h. No visual agglomerates were found in the ferrofluids with *w* less than 40 wt%; however, some obvious agglomerates appeared in the fluids with a *w* of 50 wt%. For further characterization of the stability of ferrofluids with a *w* of 40 wt%, some fluids were taken from the centrifuge tube; the TEM image is shown in Figure 3. The Zn-ferrite nanoparticles were dispersed uniformly in the base liquid PFPE oils and no agglomerations of particles appeared, as shown in the TEM image (Figure 2b); in addition, *λ*′ = 0.51 (Appendix A), indicating the excellent stability of the synthesized PFPE oil-based ferrofluids. 

### 3.3. The Rheological Properties of PFPE Oil-Based Ferrofluids

Figure 4 shows the rheological properties of PFPE oil-based ferrofluids prepared with a series of particle concentrations *w.* In a Newtonian fluid, *τ* = *ηγ*, of which *γ* is shear rate, *η* is the viscosity of the fluid and *τ* is the shear stress. *τ* is proportional to *γ*, and *η* is almost unchanged. As shown in Figure 4a, *τ* was nearly proportional to *γ* when *w* was lower than 30 wt%, indicating that the ferrofluid was a Newtonian fluid. However, the *τ–γ* relationship was not proportional and *τ* exceeded yield stress *τ*_y_ when *w* was 40 wt%, indicating that the ferrofluid was a non-Newtonian fluid. 

The viscosity of ferrofluid with a *w* of 40 wt% decreased as shear rate increased, and shear thinning appeared (Figure 4b). It has been reported that when the dipolar coupling constant *λ*′ > 1, the nanoparticles in the ferrofluids have the possibility of chain formation [52]. The size distribution of Zn-ferrite nanoparticles is in the range of 4–18 nm, there are some large nanoparticles in the ferrofluids, and when the particle size *d* is 18 nm, the *λ*′ = 2.01, indicating that some chain-like nanoparticles appear. It was reported that the chain-like structures would be broken by shear action; moreover, the arrangement number of nanoparticles reduces with increasing shear rate [53]. Besides this, the base liquid molecules in PFPE oils originally entangled with each other are stretched and arranged along the shear direction, and thus, these two factors contribute to the decrease in viscosity and the appearance of shear thinning.

As shown in Figure 4c, the viscosity of the ferrofluid increased slowly with increasing *w* when it was less than 30 wt%; however, it increased rapidly when *w* reached 40 wt%. The viscosity was much higher than that obtained from the linear relationship than what has been previously reported, due to the chain-like structures changing to be irregular [53]. Generally, too high a viscosity of ferrofluid is adverse in sealing, due to the fact that the friction and high temperature induced by viscosity will lower sealing pressure [7]. 

In addition, the activity of fluids changed to be solid-like when *w* exceeded a maximum value *w*_m_ [54]. *η*_r_ = *η*/*η*_0_, where *η*_0_ and *η* is the viscosity of base liquid and ferrofluid, respectively. The curve relationship between 1 − *η*_r_^−1/2^ and *w* of PFPE oil-based ferrofluid is fitted in Figure 4d. When *η* was close to ∞ and 1 − *η*_r_^−1/2^ was close to 1, the activity of PFPE oil-based ferrofluid changed to be solid-like, and the *w*_max_ could was 45.5 wt% (R^2^ = 99.5%). 

As mentioned above, the activity of the ferrofluids changed from Newtonian to Non-Newtonian, then to solid-like with increasing *w* from 10 wt% to 45.5 wt%, due to the increase in nanoparticle interactions. The viscosity of the ferrofluids with *w* less than 30 wt% was low, and the fluids were suitable for high rotation sealings; the fluids with *w* more than 40 wt% were suitable for low rotation sealings or big gap sealings [8]. Morever, when *w* was more than 45.5 wt%, the ferrofluids lost liquidity and could not be used in sealings.

The effect of PFPE oil polymers’ *M*_w_ on the rheological properties of ferrofluids is shown in Figure 5. It has been reported that the viscosity of polymers increases as the *M*_w_ increases [55]. When *M*_w_ was lower than 3500 g/moL, the viscosity curve tended to be horizontal with increasing shear rate, owing to the rapid arrangement of PFPE oil molecules along the shear direction (as shown in Figure 5a). However, when *M*_w_ was higher than 4600 g/moL, shear thinning appeared with increasing shear rate, owing to the fact that the polymer molecules with higher *M*_w_ firstly tangle more severely with each other, and then arrange gradually along the shear direction. In Figure 5b, the viscosity was essentially unchanged as shear time went on, which shows that the ferrofluids were dispersed uniformly and no agglomerations came into being under shear action, attributing to that the rheological properties of ferrofluids are sensitive to the microstructures [26]. The viscosity increased with increasing *M*_w_, and it increased remarkably when *M*_w_ was larger than 4600 g/moL (as shown in Figure 5c), attributed to the stronger entanglements of base liquids molecules; thus, the viscosity of base liquid PFPE oil increased with increasing *M*_w_. The PFPE oil with *M*_w_ lower than 4600 g/moL was a good choice for synthesizing ferrofluid with low viscosity. Besides this, the viscosity of the fluids under a magnetic field was higher (as shown in Figure 5d) than that of the fluids without an applied magnetic field (as shown in Figure 5a), owing to the appearance of the complex chain-like, or even column-like, arrangement of nanoparticles under a magnetic field [56].

The performance stability of ferrofluids is of great importance in various studies and applications. It has been reported that PFPE oil-based ferrofluids have temperature-resistance under the range of 0–140 °C [57]. Figure 6 shows the rheological properties of PEPE oil-based ferrofluids under various temperatures. It can be observed in Figure 6a that the viscosity of the ferrofluids reduced as temperature rose. The viscosity of PFPE oil reduces with rising temperature, as revealed by T. J. Fortin et al. [58]. Besides this, the thermal energy *kT* increases and the inter-molecular friction reduces with rising temperature in the suspensions [59]. We found that the influencing mechanisms of the viscosity temperature change of ferrofluids are more complex than what have been reported [60]. Significantly, the saturation magnetization of Zn-ferrite nanoparticle also reduced with rising temperature (Figure 2f), and thus, all these factors together contribute to the viscosity reduction in Zn-ferrite PFPE oil-based ferrofluid. In Figure 6b, shear thinning appeared due to the arrangement of particles and base liquid molecules along the shear direction. The viscosity reduced with rising temperature, showing the same tendency as that when the magnetic field was 0 mT. According to the above results, the viscosity of PFPE oil-based ferrofluids under various temperatures could be forecasted.

To our knowledge, the rheological properties of ferrofluids under a magnetic field are more complicated and difficult to describe accurately than those without an applied magnetic field [61]. The force caused by magnetic field strength on magnetic nanoparticles is obtained through the following equation [62]:
$$F_{m}=\frac{\mu_{0}M_{0}^{2}\pi d^{6}}{24{\left(d+2b\right)}^{4}}$$
where, *M*_0_ is the magnetization of nanoparticles, which it reaches saturation with increasing magnetic field, *d* is average particle size, *b* is the shell thickness of coated nanoparticles, and *µ*_0_ is the permeability of the vacuum. According to the equation, the caused force *F*_m_ by magnetic field strength on magnetic nanoparticles increases with increasing *M*_0_^2^; meanwhile, the particles arrange to be chain-like structures, columns, or even drop-like, closed rings and branched structures along the magnetic field direction.

The force caused by the shear rate on nanoparticles is obtained through the following equation [62]:
$$F_{v}=6\pi \eta_{c}\gamma \frac{1}{2}{\left[\frac{N(d+2b)}{2}\right]}^{2}$$
where, *N* is the number of nanoparticles in chain structures and *η*_c_ is the viscosity of the base liquid. According to the equation, the caused force *F*_v_ by the shear rate on nanoparticles increases with increasing shear rate *γ*, and meanwhile, the nanoparticles and base liquids molecules arrange along the shear direction.

The rheological properties of PFPE oil-based ferrofluid under a magnetic field are shown in Figure 7. When shear rate was lower than 10 s^−1^, shear thickening appeared, and the viscosity curve increased first and then tended to be horizontal (at ~200 mT) with increasing magnetic field strength. It has been known that a chain arrangement of nanoparticles will appear in synthesized PFPE oil-based ferrofluids, owing to the fact that the dipolar coupling constant *λ*′ = 2.0 (SI) for some large particles. The chain arrangement of nanoparticles will change into columns as the magnetic field strength increases, and the change in microstructures contributes to the viscosity change [63]. According to Figure 2e, the saturation magnetization of Zn-ferrite nanoparticles increased first and then tended to be horizontal as magnetic field strength increased. Thus, the curve increment was due to that the arrangement number of microstructures gradually increased, and it was difficult for shear action to destroy and rearrange the microstructures. The curve horizontal segment was due to the fact that the number of microstructures reached saturation, and the shear action tended to be stable. However, when the rate was higher than 50 s^−1^, the curves tended to be horizontal and viscosity was almost unchanged (Figure 7a), contributing to the magnetization of particles reaching saturated, and the chain-like and column structures basically remaining unchanged. Besides this, the viscosity reduced with increasing shear rate, and the increase in shear rate was favorable for the rearrangement of micro-structures. In general, the viscosity-magnetization change was contributed by the magnetic field and shear rate.

The viscosity of the ferrofluid reduces with increasing shear rate (Figure 7b), attributed to the fact that the nanoparticle and base liquid molecules rearrange under shear action. Besides this, the viscosity increased as magnetic field strength increased, and the nanoparticles re-arranged from chain-likes to columns. Meanwhile, it was more difficult to destroy and rearrange the microstructures along the shear direction. When *w* was less than 30 wt%, the viscosity increased first and then tended to be level with increasing magnetic field strength (Figure 7c). However, when *w* was 40 wt%, the viscosity increased slowly and fluctuated, attributed to the fact that some nanoparticles aggregated to form larger clusters, and because the instability of the ferrofluid appeared. The viscosity increased with increasing *w* at 10 mT (Figure 7d), revealing the same tendency, but larger than that without an applied magnetic field (Figure 3)—attributed to the arrangement of microstructures under the magnetic field and shear action.

As mentioned above, the viscosity and its alteration in Zn/ferrite PFPE oil-based ferrofluids will be affected by multiple factors, which can be predicted under various shear rates, temperatures and magnetic fields; this is important for all kinds of applications, especially in different types of sealings. 

## 4. Conclusions

In this study, the Zn-ferrite nanoparticles were chemically coated by PFPE acids and equably dispersed in PFPE oil. For maintaining liquidity of the ferrofluids, *w* should be less than 45.5 wt%. For avoiding too much friction caused by high viscosity and instability of the ferrofluids, *w* should be less than 30 wt% and *M*_w_ should be less than 4600 g/moL. The viscosity of the ferrofluids decreases with rising temperature. The viscosity is mainly affected by shear rate when it is higher than 50 s^−1^, and the viscosity is mainly affected by magnetic field strength when it is less than ~200 mT and when shear rate is less than 10 s^−1^. The arrangement of nanoparticles and base liquid molecules are together affected by shear rate and magnetic field strength. According to the obtained rheological properties of Zn-ferrite/PFPE oil-based ferrofluids, the viscosity and changes in it could be predicted when the fluids were applied under various temperatures, shear rates and magnetic field strengths.

## Figures and Tables

**Figure 1 nanomaterials-11-02653-f001:**
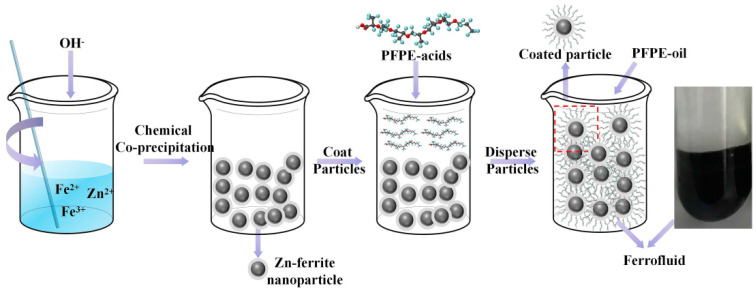
Schematic illustration of the preparation process for Zn-ferrite nanoparticles and PFPE oil-based ferrofluid.

**Figure 2 nanomaterials-11-02653-f002:**
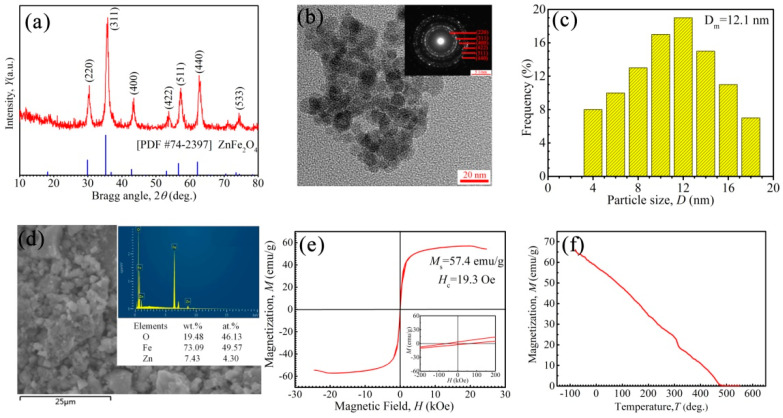
(**a**) XRD pattern, (**b**) TEM image, (**c**) the particle size distribution and (**d**) the SEM and EDS of Zn-ferrite nanoparticle (Zn_*x*_Fe_3−*x*_O_4_, the ratio of *x* is 0.2), (**e**) VSM curve of coated Zn-ferrite nanoparticles at 25 °C, and (**f**) the magnetization of Zn-ferrite nanoparticles vs. temperature (3 T).

**Figure 3 nanomaterials-11-02653-f003:**
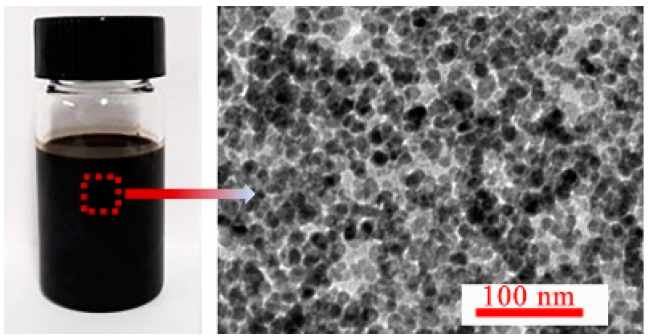
The TEM image of PFPE oil-based ferrofluids with *w* of 40wt%.

**Figure 4 nanomaterials-11-02653-f004:**
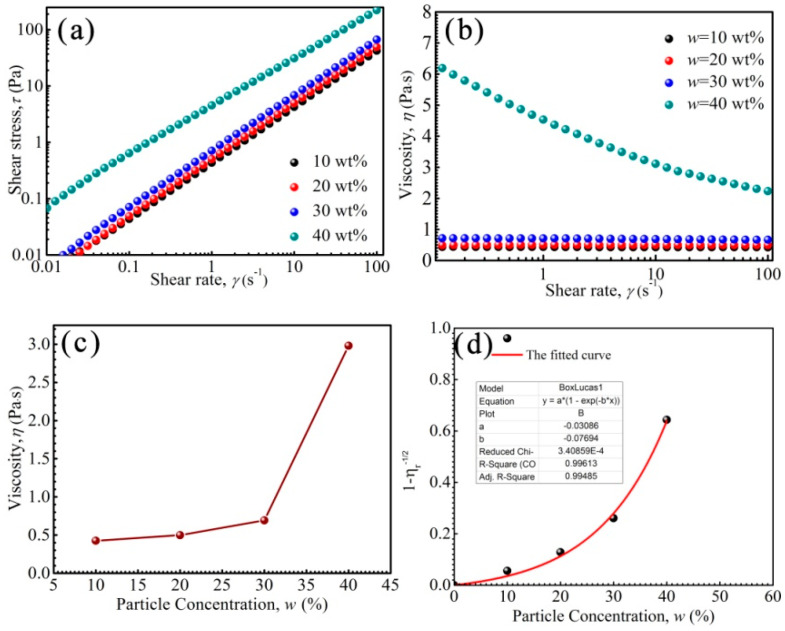
The shear stress of the ferrofluids as a function of shear rate (**a**), the viscosity as a function of shear rate (**b**), the viscosity as a function of nanoparticle concentrations *w* (**c**) and the 1 − *η*_r_^−1/2^ as a function of *w* (**d**).

**Figure 5 nanomaterials-11-02653-f005:**
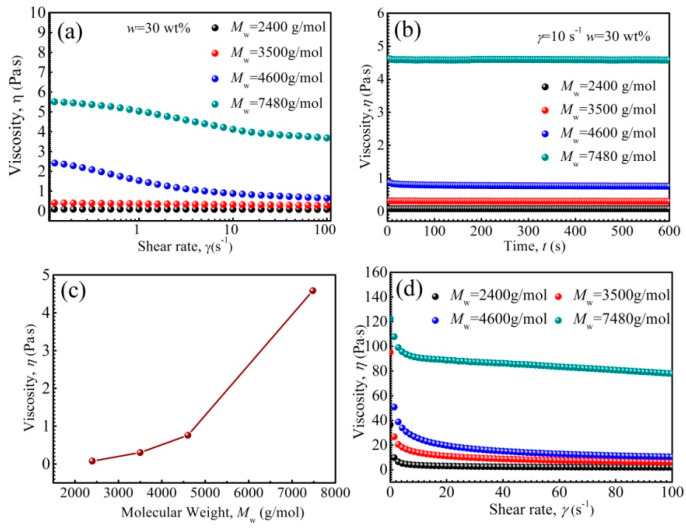
The viscosity of ferrofluids as a function of shear rate (**a**), shear time (**b**), PFPE oil *M*_w_ (**c**), and the viscosity of the fluids as a function of shear rate at 10 mT (**d**).

**Figure 6 nanomaterials-11-02653-f006:**
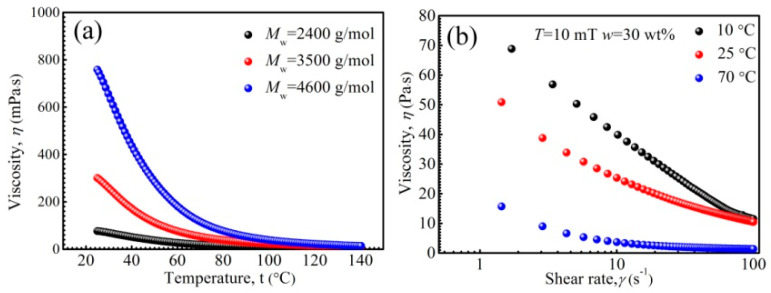
The viscosity of ferrofluids as a function of temperature (**a**), the viscosity of ferrofluids as a function of shear rate (**b**).

**Figure 7 nanomaterials-11-02653-f007:**
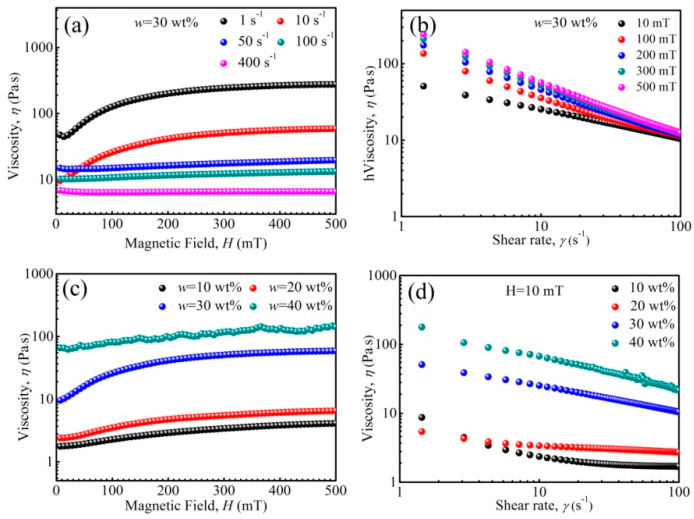
The viscosity of ferrofluids as a function of magnetic field (**a**,**c**), and shear rate (**b**,**d**).

## Data Availability

Not applicable.

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
