# Peer review of "Investigation of the Rheological Properties of Zn-Ferrite/Perfluoropolyether Oil-Based Ferrofluids"

_nanomaterials, 2021, doi:10.3390/nano11102653_

Round 1

Reviewer 1 Report

The manuscript refers to the synthesis and rheological properties of perfluoropolyether-oil (PFPE-oil) based ferrofluids, a type of magnetic fluids very useful in rotating seal applications.

Comments.

  1. 2, part of Introduction, is sometimes confusing. Beside the carrier liquid properties, the ferrofluid colloidal stability and particle volume fraction have the most important influence on the flow properties. E.g., a weak colloidal stability induces particle agglomerate formation, which has a major effect on the effective viscosity. In this respect there have to be take into account the efficiency of stabilization mechanism and the particle volume fraction, beside the dipolar coupling constant, factors heavily influencing agglomerate formation. If water-based ferrofluids are discussed for comparative reasons, it is important to specify the type of stabilization procedure applied: electrostatic or electro-steric.

Essentially, in case of ferrofluids designed for rotating seal applications the magnetoviscous effect should be reduced, which means negligible agglomerate formation.

Ref. [29] refers to magnetorheological fluids, not ferrofluids.

It is recommended to fully reconsider the introductory text.

  1. The value of the dipolar coupling constant λ is well above 1, therefore the ferrofluid is a strongly aggregating one. This is well evidenced especially by the magnetorheological properties, as the authors have shown. In this respect, the magnetoviscous effect should be correctly represented as the relative increase of effective viscosity vs. magnetic induction (T) and the relative increase of effective viscosity vs. particle volume fraction. I suggest to avoid mass concentration dependences and also mass magnetization in case of ferrofluids.

Minor comments

  1. Introduce also the titles of references:
  2. Chen; X. Yang; S. Gao. Int. J. Appl Electrom, 2017, 55, 535
  3. Bădescu; D. Condurache; M. Ivanoiu. J. Magn. Magn. Mater., 1999, 202,197-200 (verify also year/vol/pages)

Author Response

Comments

  1. Q:2, part of Introduction, is sometimes confusing. Beside the carrier liquid properties, the ferrofluid colloidal stability and particle volume fraction have the most important influence on the flow properties. E.g., a weak colloidal stability induces particle agglomerate formation, which has a major effect on the effective viscosity. In this respect there have to be take into account the efficiency of stabilization mechanism and the particle volume fraction, beside the dipolar coupling constant, factors heavily influencing agglomerate formation.

A:Thank you so much for your suggestions. The introduction has been modified. It is known that the rheological properties of ferrofluids are affected by many factors. Here, the factors are divided into two categories, one category is shear conditions, such as shear rate, temperature and magnetic field strength; another category is the differences of the ferrofluids, such as the types of the base liquids, the nanoparticles concentration and nanoparticle size.

Q: If water-based ferrofluids are discussed for comparative reasons, it is important to specify the type of stabilization procedure applied: electrostatic or electro-steric.

A: Thank you for your advice. The water based ferrofluids related references have been replaced.

Q: Essentially, in case of ferrofluids designed for rotating seal applications the magnetoviscous effect should be reduced, which means negligible agglomerate formation.

A: The increase of nanoparticle concentration will promote the saturation magnetization of ferrofluids and will further improve sealing pressure, which is important in big gap sealings or in high pressure gap sealings. Sometimes the requirement of dispersion stability of ferrofluids will be reduced.

Q: Ref. [29] refers to magnetorheological fluids, not ferrofluids.

A: Thanks a lot, the reference has been replaced.

Q: It is recommended to fully reconsider the introductory text.

A: Thanks a lot. The introductory has been modified. In this paper, the PFPE-oil based ferrofluids are often used in various sealings, thus, sometimes our perspectives are often related to the industrial applications.

  1. Q: The value of the dipolar coupling constant λ is well above 1, therefore the ferrofluid is a strongly aggregating one. This is well evidenced especially by the magnetorheological properties, as the authors have shown.

A: λ’ has been re-calculated, it is 0.51 and less than 1, indicating the well  dispersion of stability of the ferrofluids, and the detailed calculation process is shown in SI.

Q: In this respect, the magnetoviscous effect should be correctly represented as the relative increase of effective viscosity vs. magnetic induction (T) and the relative increase of effective viscosity vs. particle volume fraction. I suggest to avoid mass concentration dependences and also mass magnetization in case of ferrofluids.

A: You are right. The effective viscosity vs. magnetic induction and the effective viscosity vs. particle volume fraction are usually expressed by most authors. However, the relationship between viscosity and the mass fraction are more direct, and we can obtain the ferrofludis with various mass fraction just by controlling the experiment ratio. And the viscosity vs. volume fraction can be obtained by formulas, thus the mass  fraction is converted to the volume fraction in SI.

Besides, in industrial applications, the viscosity and not effective viscosity is more frequently used.

Minor comments

  1. Introduce also the titles of references:
  2. Chen; X. Yang; S. Gao. Int. J. Appl Electrom, 2017, 55, 535
  3. X. Long, L. D. Cai. Experimental investigation of diverging stepped magnetic fluid seals with large sealing gap[J]. Int. J. Appl Electrom, 2016, 50(3):407-415.
  4. Bădescu; D. Condurache; M. Ivanoiu. J. Magn. Magn. Mater., 1999, 202,197-200 (verify also year/vol/pages)
  5. Bădescu, D. Condurache, M. Ivanoiu. Ferrofluid with modified stabilisant[J]. J. Magn. Magn. Mater., 1999, 202(1):197-200.

Reviewer 2 Report

The manuscript by Fang Chen and co-authors reports the experimental studies of the rheological behavior of the Zn-ferrite/perfluoropolyether-oil (PfPE-oil) based ferrofluids. Complex measurements were performed varying the particle concentration, the flow shear rate, and the applied magnetic field strength. The present experimental data reveal very interesting physical effects due to structural behavior of magnetic nanoparticles suspended in the ferrofluids. The research material is perfectly in line with the topics of interest of Nanomaterials, and it deserves publication I n Nanomaterials.

At the same time that I am delighted with the experimental results, I am completely dissatisfied with the interpretation.

Probably all peculiarities in the rheological behavior of ferrofluids are originated by the structural transformation in an ensemble of ferroparticles. The bibliography on this topic is very wide, and the authors cite it very limited. Taking into account the magnetic field effects, two main structural transformations are worth mentioning. The first one is the magnetic field induced phase separation resulting in formation of micron-sized drop-like aggregates. The effect is well described, for example, in a recent review Materials 2020 (13) 3956. The second one is the chain-aggregate formation, including their lengthening under the presence of a magnetic field [for example, Physical Review E 2004 (70) 051502]. So, on my opinion the Bibliography should be extended.

The viscosity of ferrofluids increases strongly with particle concentration; it is well-known from many experiments [for example Physical Chemistry Chemical Physics 2016 (18) 18342] that this dependence exceeds the linear one, as it is shown in Fig. 4. But it is also clear from Fig. 4d, that linear line is poor here, and the law is parabolic at least. Thus the loss of fluidity is at the mass concentration 42-45 wt%, and this estimate is close to the volume fraction 0.5-0.6 (Suppl. 1).

It is absolutely clear that the Sample with 40 wt% demonstrates the different properties from other samples. This point should be investigated and discussed in the paper.

Suppl. 2. The particle magnetic moment is independent on the ferrofluid concentration; it is the feature of the nanoparticle. So, the particle magnetic moment is dependent on the bulk magnetization of the ferroparticle material. And the ferrofluid magnetization saturation is proportional to the mean magnetic moment of the particles, since they are distributed over sizes. All estimations here are not correct.

Suppl. 3. What is the definition of dipolar coupling constant here? Commonly, it contains the cubic power of hydrodynamic diameter of the particle. The value 71 is fantastic! With the interparticle interaction of this STRENGTH, all particles must be aggregated in one VERY BIG aggregate!

In addition, \lambda is used twice, for the dipolar coupling and for the X-ray wave length.

I support the paper, but I'd like to ask the authors for major revision.

Author Response

At the same time that I am delighted with the experimental results, I am completely dissatisfied with the interpretation.

Q: Probably all peculiarities in the rheological behavior of ferrofluids are originated by the structural transformation in an ensemble of ferroparticles. The bibliography on this topic is very wide, and the authors cite it very limited. Taking into account the magnetic field effects, two main structural transformations are worth mentioning. The first one is the magnetic field induced phase separation resulting in formation of micron-sized drop-like aggregates. The effect is well described, for example, in a recent review Materials 2020 (13) 3956.The second one is the chain-aggregate formation, including their lengthening under the presence of a magnetic field [for example,  ]. So, on my opinion the Bibliography should be extended.

A: Thank you very much for your suggestions. The references have been supplemented, especially the drop-like aggregates and chain-aggregate formation related references.

Q: The viscosity of ferrofluids increases strongly with particle concentration; it is well-known from many experiments [for example Physical Chemistry Chemical Physics 2016 (18) 18342] that this dependence exceeds the linear one, as it is shown in Fig. 4. But it is also clear from Fig. 4d, that linear line is poor here, and the law is parabolic at least. Thus the loss of fluidity is at the mass concentration 42-45 wt%, and this estimate is close to the volume fraction 0.5-0.6 (Suppl. 1).

A: Only when the nanoparticle concentration is very low, even far lower than 5wt%, the ferrofluids are rigorous Newtonian fluid. When w is 40wt%, the ferrofluids are Non-newtonian fluid, the ferrofluids with w of 10wt% to 30wt% are taken as Newtonian fluid, due to that the τ is basically proportional to γ.

Q: It is absolutely clear that the Sample with 40 wt% demonstrates the different properties from other samples. This point should be investigated and discussed in the paper.

A: When w is 40wt%, the obvious differences appear. In fact, the dispersion stability of the ferrofluids reduces with increasing nanoparticles concentration. When shear action occours, the obvious fluctuation (Fig. 7(c)) appears, due to the agglomeration of particles occurs and the chain-like structures change to be irregular.

Suppl. 2. The particle magnetic moment is independent on the ferrofluid concentration; it is the feature of the nanoparticle. So, the particle magnetic moment is dependent on the bulk magnetization of the ferroparticle material. And the ferrofluid magnetization saturation is proportional to the mean magnetic moment of the particles, since they are distributed over sizes. All estimations here are not correct.

A: The detailed calculation process is shown as below.

Suppl. 3.  Q: What is the definition of dipolar coupling constant here?

A: The interparticle interaction in ferrofluids is expressed by the interaction parameter (λ’) as , in which  is the vacuum permeability,  is the spontaneous magnetization of the magnetic material,  is its volume, d is the particle diameter,  is the Boltzmann constant, and T is the absolute temperature. In the formula, =H/m,  is the bulk magnetization of the nanoparticle, it can be calculated according to = 57.4 A·m2/kg×5.18×103 kg/m3=2.97×105 A/m, , in which, dTEM=12.1nm, d= dTEM +2s=12.1nm+2×3.6nm=19.3nm, in which, s is the thickness of the coated surfactant, according to reference [42], =1.3810-23J/K, T=298K. λ’ is calculated as 0.51, which is less than 1, indicating the well dispersion stability of the ferrofluids.

The value has been calculated for many times and by several people.

Q: Commonly, it contains the cubic power of hydrodynamic diameter of the particle. The value 71 is fantastic! With the interparticle interaction of this  STRENGTH, all particles must be aggregated in one VERY BIG aggregate!

A: The right value of  is 0.51. Thank you for your guidance on the understand of various parameters in the formula.

Q: In addition, \lambda is used twice, for the dipolar coupling and for the X-ray wave length.

A: Thank you for your reminding, and the dipolar coupling is denoted by λ’.

Round 2

Reviewer 1 Report

The manuscript is still inadequate for publication.

An example of many is on pg.9: " µ0 is the permeability of nanoparticles", a big mistake! Actually this is the permeability of vacuum.

Author Response

Thank you so much for your advices.

 Q:The manuscript is still inadequate for publication.

A:The paper has been extensively modified this time.

Q:An example of many is on pg.9: " µ0 is the permeability of nanoparticles", a big mistake! Actually this is the permeability of vacuum.

A: The mistake has been modified.

Reviewer 2 Report

In my first review I asked the authors to improve strongly the physical interpretation and discussion of the obtained experimental results, as well as to extend the bibliography. Unfortunately, I have to state that the authors have not shown a sincere desire to revise and improve the article. Actually, the revision appeared to be limited with two paragraphs at page 2 (and the addition of several references) and the recalculation of Suppl. 2.

Description of magnetoviscous phenomena in ferrofluids needs to be accompanied by the qualitative explanation of the physical effects. And the used citations must not be taken in some random way. For example, Lines 68-69: “… and revealing the chains formation along the magnetic field direction. Ref. [32]”. What is Ref. [32] here? It is written with a number of mistakes, and this paper is devoted to technical application!? The following references are more appropriate here: for example, Physical Review E 2004 (70) 051502 (theory); Journal of Physics Condensed Matter 2008 (20) 204113 (experimental verification with CRYO-TEM); and many others. In addition, it is worth mentioning that the intensive magneto-dipole interaction (\lambda’ ~ 7 - 8) results in more complication clusters than linear chains, see, for example, Physical Chemistry Chemical Physics 2015 (17) 16601 and THE JOURNAL OF CHEMICAL PHYSICS 139, 134901 (2013).

Suppl. 2: The dipolar coupling constant is not the artificial parameter inserted in theoretical models or in computer simulations. This parameter is of major importance because it measures the intensity of the interparticle magneto-dipole interaction in comparison to thermal energy. In the first version of the paper this parameter was given as 71. With this intensity all ferroparticles must be clustered in very big aggregates, and the suspension cannot be colloidal stable. In the present second version this parameter is calculated as 0,51. In other words, the dipolar interaction is very weak; and the particles cannot form the chain-like aggregates due to thermal motion. So, the extensive discourse on chain formation does not make sense. And the statements, given below, are incorrect:

Line 286: “…the dipolar coupling constant is very big and λ’=0.51 (S1).”

Lines 194 – 197: “… It has been reported that when the dipolar coupling constant λ’=8, the long-chains of nanoparticles formed even without applied magnetic field. The λ’ of the synthesized PFPE-oil based ferrofluid is calculated and obtained as λ’=0.51 (S1), indicating the chains formation of Zn-ferrite nanoparticles. “

The authors did not pay attention to my remark concerning Fig. 4d (!?) Drawing the red straight line here results in 62 wt% solid-like concentration. At the same time, Table S1 shows that the 40 wt% concentration is equal to 50 % of the particle volume fraction. Hence, the 62 wt% is equal to 75 % of volume concentration, and this value seems to be unachievable. In my previous review I suggested to approximate the experimental dots with the parabolic line, which will give the threshold value of 42-45 wt%, and 50-55 % of volume fraction (Suppl. 1). I do not understand why the authors ignored this remark, presenting unphysical numbers inconsistent with each other.

Concluding, the manuscript was not revised well; and the second version does not satisfy the high-quality level of Nanomaterials.

Author Response

Comments and Suggestions for Authors

Q:In my first review I asked the authors to improve strongly the physical interpretation and discussion of the obtained experimental results, as well as to extend the bibliography. Unfortunately, I have to state that the authors have not shown a sincere desire to revise and improve the article. Actually, the revision appeared to be limited with two paragraphs at page 2 (and the addition of several references) and the recalculation of Suppl. 2.

Description of magnetoviscous phenomena in ferrofluids needs to be accompanied by the qualitative explanation of the physical effects. And the used citations must not be taken in some random way. For example, Lines 68-69: “… and revealing the chains formation along the magnetic field direction. Ref. [32]”. What is Ref. [32] here? It is written with a number of mistakes, and this paper is devoted to technical application!?

The following references are more appropriate here: for example, Physical Review E 2004 (70) 051502 (theory); Journal of Physics Condensed Matter 2008 (20) 204113 (experimental verification with CRYO-TEM); and many others. In addition, it is worth mentioning that the intensive magneto-dipole interaction (\lambda’ ~ 7 - 8) results in more complication clusters than linear chains, see, for example, Physical Chemistry Chemical Physics 2015 (17) 16601and THE JOURNAL OF CHEMICAL PHYSICS 139, 134901 (2013).

A:The paper has been extensively modified, most of which is highlighted in red.

Q:Suppl. 2: The dipolar coupling constant is not the artificial parameter inserted in theoretical models or in computer simulations. This parameter is of major importance because it measures the intensity of the interparticle magneto-dipole interaction in comparison to thermal energy. In the first version of the paper this parameter was given as 71. With this intensity all ferroparticles must be clustered in very big aggregates, and the suspension cannot be colloidal stable. In the present second version this parameter is calculated as 0,51. In other words, the dipolar interaction is very weak; and the particles cannot form the chain-like aggregates due to thermal motion. So, the extensive discourse on chain formation does not make sense. And the statements, given below, are incorrect:

 Line 286: “…the dipolar coupling constant is very big and λ’=0.51 (S1).”

Lines 194 – 197: “… It has been reported that when the dipolar coupling constant λ’=8, the long-chains of nanoparticles formed even without applied magnetic field. The λ’ of the synthesized PFPE-oil based ferrofluid is calculated and obtained as λ’=0.51 (S1), indicating the chains formation of Zn-ferrite nanoparticles.

A:Thank you very much for your patient guidance and I have learned a lot from you.

λ’=0.51 (S1), indicating the well dispersion stability of ferrofluids.

However, it was reported that when the particle size is bigger than 16nm, the chainlike structures appear. Since the dipole-dipole energy increases with the size of particles, only ‘large’ particles can form structures.

In my paper, the size distribution span of the Zn-ferrite is from 4 nm to 18 nm. And when the dTEM is 18nm, the λ’=2.0, indicating that some chainlike structures appearance in the ferrofluids.

Besides, it was reported that the real magnetic fluids are practically always polydisperse and often contain ferromagnetic particles that are large enough to agglomerate. The biggest particles in real ferrofluids can play a principal role in formation of rheological properties of these systems in spite of their small concentration.

What’s more, a strong dependence of the scattering intensity on the volume concentration of magnetic particles has been observed by SANS, indicating for highly concentrated ferrofluids, even at zero field, cluster formation due to magnetic dipole-dipole interaction. In our paper, all the ferrofluids are dense.

Q:The authors did not pay attention to my remark concerning Fig. 4d (!?)Drawing the red straight line here results in 62 wt% solid-like concentration. 62wt% At the same time, Table S1 shows that the 40 wt% concentration is equal to 50 % of the particle volume fraction. Hence, the 62 wt% is equal to 75 % of volume concentration, and this value seems to be unachievable. In my previous review I suggested to approximate the experimental dots with the parabolic line, which will give the threshold value of 42-45 wt%, and 50-55 % of volume fraction (Suppl. 1). I do not understand why the authors ignored this remark, presenting unphysical numbers inconsistent with each other.

A:Thank you for your guidance. wm is obtained as 45.5 wt%. The experiment dots are fitted by a new formula, and R2=99.5.

Concluding, the manuscript was not revised well; and the second version does not satisfy the high-quality level of Nanomaterials.

Round 3

Reviewer 2 Report

I recommend the revised version 3 for publication in Nanomaterials/

Author Response

Thanks a million for your kind suggestions. The corrections of the manscripts has been signed as coloured words.
